# Generating a Cylindrical Panorama from a Forward-Looking Borehole Video for Borehole Condition Analysis

**Zhaopeng Deng** [1] , **Maoyong Cao** [1,2,*] , **Yushui Geng** [2] **and Laxmisha Rai** [1]

[1] College of Electrical Engineering and Automation, Shandong University of Science and Technology, Qingdao 266590, Shandong, China

[2] College of Electrical Engineering and Automation, Qilu University of Technology, Jinan 250353, Shandong, China

* Correspondence: my-cao@263.net

**Abstract:** Geological exploration plays a fundamental and crucial role in geological engineering. The most frequently used method is to obtain borehole videos using an axial view borehole camera system (AVBCS) in a pre-drilled borehole. This approach to surveying the internal structure of a borehole is based on the video playback and video screenshot analysis. One of the drawbacks of AVBCS is that it provides only a qualitative description of borehole information with a forward-looking borehole video, but quantitative analysis of the borehole data, such as the width and dip angle of fracture, are unavailable. In this paper, we proposed a new approach to create a whole borehole-wall cylindrical panorama from the borehole video acquired by AVBCS, which provides a possibility for further analysis of borehole information. Firstly, based on the Otsu and region labeling algorithms, a borehole center location algorithm is proposed to extract the borehole center of each video image automatically. Afterwards, based on coordinate mapping (CM), a virtual coordinate graph (VCG) is designed in the unwrapping process of the front view borehole-wall image sequence, generating the corresponding unfolded image sequence and reducing the computational cost. Subsequently, based on the sum of absolute difference (SAD), a projection transformation SAD (PTSAD), which considers the gray level similarity of candidate images, is proposed to achieve the matching of the unfolded image sequence. Finally, an image filtering module is introduced to filter the invalid frames and the remaining frames are stitched into a complete cylindrical panorama. Experiments on two real-world borehole videos demonstrate that the proposed method can generate panoramic borehole-wall unfolded images from videos with satisfying visual effect for follow up geological condition analysis. From the resulting image, borehole information, including the rock mechanical properties, distribution and width of fracture, fault distribution and seam thickness, can be further obtained and analyzed.

**Keywords:** borehole video; video image mosaic; SAD; gray projection transformation

## 1. Introduction

Borehole camera technology (BCT) is an important optical signal acquisition and processing technology to image the interior structure of boreholes. It has been widely used in geological detection [1], investigating the structure of glaciers [2] and hydrogeological investigation [3]. Among BCT techniques, the digital panoramic borehole camera system (DPBCS) [4] and axial view borehole camera system (AVBCS) [5] are the two most widely used methods. The principle of a DPBCS device is a conic mirror installed on a probe to map the 3D borehole-wall image to a 2D plane image and

thus a borehole-wall cylindrical panorama can be produced by image mosaic technology. However, this complex and expensive equipment can only be applied to vertical drilling and the diameter for the test borehole should be in the range from 48 to 180 mm. In comparison, the AVBCS device is relatively simple and cheap. Significantly, it can be applied to the vertical, horizontal and oblique holes and can also be applied in wider drilled holes whose diameters range from 28 mm to 250 mm. The operation principle of an AVBCS is that it takes forward-looking video while the probe moves along the pre-drilled borehole in an axial motion; engineers, by studying the playback or screenshot of the borehole video, can obtain data on the geological structure. However, AVBCS only allows qualitative analysis from the videos. Quantitative data, which will be applied in the analysis of the fracture, joint, structural plane, etc., are not available. Therefore, it is of great practical significance to create a method to generate a panoramic cylindrical image from the borehole video obtained by AVBCS. In addition, some experts use an introscopic camera to take photographs to analyze the rock mass characteristics in boreholes. Table 1 shows the main parameters of the AVBCS used in this study, DPBCS and the introscopic camera used in the literature [6].

**Table 1.** Technical specifications of different devices.

| Device | Probe Diameter (mm) | CCD Pixels | Color (Bits) | Borehole Diameter (mm) | Test Velocity (m/min) |
|---|---|---|---|---|---|
| DPBCS [4] | 45 | $795 \times 596$ | 16/24 | 48–180 | 1.5 |
| AVBCS [5] | 25 | $320 \times 240$ | 16/24 | 28–250 | 0.8 |
| Introscopic camera [6] | 50 | $1280 \times 720$ | 16/24 | >60 | \ |

Although the video mosaic technique has been deeply investigated for several years, it is still challenging in the field of machine vision. Videos are generally taken in two ways: One is by rotating a camera around its optical axis in a fixed position and the other is to take videos with a moving camera [7]. The process of obtaining a borehole video with AVBCS should be classified as a monocular moving camera. To be more specific, a single CCD (charge-coupled device) camera is installed on the top of the probe, and, accompanied with slight rotation and vibration, the probe moves forward along the central axis of a drilled hole. A literature review shows that there are few studies on borehole video processing with AVBCS, and even fewer on generating cylindrical panoramas from these videos. However, there is some research on similar techniques. On pipeline inspection with mobile cameras [8,9], to which borehole video-taking is similar, there are some studies: Rousso et al. [10] proposed a manifold projection algorithm which transforms a video sequence into a parallel image sequence and thus a video mosaic can be obtained for the most challenging cases of forward motion and zoom. In [11], a method of cylindrical panoramic mosaic from a pipeline video was proposed. A series of standard circles were extracted from video sequences and converted into corresponding strips to construct a panoramic image. These studies shed light on problems in the concerned fields, but in the method proposed by Rousso, precise camera parameters have to be known beforehand and are hard to achieve with AVBCS devices; and the second study mentioned above assumes that there is no probe jitter; however, it is known that jitter exists and if the camera trajectory is not moving along the central axis of a cylinder, distortion will appear in the panorama.

The current method is proposed to solve these problems. Camera movement in a borehole is non-uniform, image distortion will arise if traditional video mosaic methods are employed, and this distortion will be more severe when probe jitter exists. Therefore, we propose a new approach to generate a wide-view borehole-wall cylindrical panorama from the video taken by AVBCS, converting the forward motion to translational motion in respect of the motion vector estimation. In the first step of this method, the drilled hole center of front view images acquired from borehole videos will be automatically positioned. The purpose is to decrease the image geometric distortion caused by camera vibration and to prepare the unfolding center for the next unwrapping process. The video image sequences are then unfolded into the corresponding rectangular image sequences by the proposed video sequence unwrapping algorithm. Finally, after invalid frames are removed, the rectangular

images are used in the image mosaic process, and with the proposed projection transformation sum of the absolute difference (PTSAD) algorithm, a single wide-view borehole-wall unfolded image will be produced.

Innovations of our work are as follows: (1) The proposed method created the complete borehole-wall cylindrical panorama from forward-looking borehole videos, which expands the application fields of the AVBCS system. This may also inspire technical innovations of pipeline inspection, nondestructive evaluation of gun muzzles, image analysis of medical endoscopic video and so on. (2) The drilled hole center of the borehole-wall image is automatically located by the proposed borehole center location algorithm, which eliminates the impact caused by the camera trajectory not being along the central axis of a drilled hole. Furthermore, the proposed virtual coordinate graph (VCG) is applied together with the coordinate mapping algorithm (CM) to construct the rectangular image sequence from the forward-looking borehole video. (3) The matching of the rectangular image sequence is the critical part of generating a complete borehole-wall cylindrical panorama. To enhance the efficiency and accuracy of the video image matching, we propose a template matching algorithm based on the sum of absolute difference (SAD) and gray projection transformation (GPT), which is referred to as the projection transformation SAD (PTSAD) algorithm.

The paper is organized as follows: The related work is reviewed in Section 2 and an overview of system descriptions is described in Section 3. Section 4 gives the description of the approach to the center location of borehole-wall images, the generation of borehole-wall unfolded images and the mosaic of borehole video images based on the proposed PTSAD algorithm. The experimental results and analysis are shown in Section 5, in which our proposed method is tested using actual borehole videos. Section 6 presents the conclusion.

## 2. Related Work

To stitch an integral panorama from a borehole video acquired by AVBCS, the front view borehole-wall image sequences obtained from the videos are first unfolded to the rectangular image sequences by the image unwrapping module. There are three basic unwrapping methods: The ray-tracing algorithm (RT) [12], look-up tables algorithm (LUTs) and the coordinate mapping algorithm (CM). In the literature [13], an image unwrapping method for spherical omnidirectional images was proposed with the RT method. However, the camera needs to be calibrated to obtain the parameters describing the spherical omnidirectional sensor in their method. Chen et al. [14] adopted the LUTs algorithm to unfold the catadioptric omnidirectional images and Dwivedi et al. [15] used decimal encoding and LUTs to create the iris template. However, the LUTs algorithm is built on the RT algorithm, which needs complex mapping libraries. In contrast, the CM algorithm is simple and efficient, and is widely used in polar-to-rectangular image transformation. Zhang et al. [16] applied the CM algorithm to a panoramic image with a visual field of 360 degrees to generate an unwrapped image. Their camera can obtain a lossless panoramic image by a parabolic reflector, thus the CM algorithm is used as a one-to-one mapping from the panoramic image to the unwrapped image. However, the front view image obtained by AVBCS in this study is a distorted image and the CM algorithm cannot be used directly. Therefore, in this paper, we introduce the interpolation algorithm to the CM algorithm to fill the missing pixels in the image unwrapping process.

After the rectangular image sequences are generated, they are stitched together as one single cylindrical panorama. The image matching of consecutive frames is a critical step in the mosaic process, which can be broadly categorized into feature-based matching and pixel intensity-based matching [17]. The feature-based methods such as Harris [18], SIFT or SURF [19] assume that distinct feature points in both the source image and the examined image can be reliably detected and elaborated by image descriptors. Given enough potent matches of feature points, it is possible to estimate the matching parameters between two images. These methods assume that the same feature points can be detected and paired successfully. However, low contrast, blur and low texture lead to a significant amount of

matching ambiguity. Furthermore, the processing speed of these methods is hard to meet the real-time processing at the video frame rate.

On the other hand, pixel intensity-based algorithms can be very robust even in low contrast images, and the template matching algorithm (TM) is the most commonly used algorithm in video image mosaics [20]. Since the TM has the advantages of simple calculation and requiring less information of the template, it has been widely used in image mosaics, pattern recognition, object tracking and so on [21]. In general, the TM involves shifting a template obtained from a source image over a search area in an examined image, measuring the similarity between the template and the current search area, and identifying the best candidate image, which matches the template most. As similarity measure methods, the normalized cross correlation (NCC) [22] and sum of absolute difference (SAD) [23] are the two most well-known and widely used in TM applications. Although NCC is more robust than SAD under variable illumination conditions, it needs to compute the numerator and denominator, which is more time-consuming than SAD. SAD is widely applied to video image matching due to its simplicity.

In order to improve the efficiency of template matching, some researchers proposed several new search strategies for SAD. Chung et al. [24] proposed an intellectual switching strategy by setting a fixed threshold for SAD to select the spatial correlation-based scheme or the temporal prediction scheme to maximize the quality improvement of the marked mosaic videos. Zhang et al. [25] set a threshold to filter the invalid candidate images in the SAD template matching. However, the adjacent frames of the borehole videos are mostly affected by certain changes such as lightness, zoom and angle. This will lead to situations where setting a fixed threshold for all video frames cannot extract the suitable candidates well. Moreover, due to the influence of these changes between templates and examined images, the position of the candidate image with the minimum SAD value is usually located at a false matching position in a number of borehole-wall images without obvious features.

For these reasons, a PTSAD algorithm is proposed for the image matching of borehole video sequences, which uses the GPT algorithm to optimize the SAD algorithm. Three candidate images with a minimum SAD value within the search region are extracted, and then the candidate image matching with the template is identified from these three candidate images by GPT. Since the GPT process reduces a 2D image to a 1D vector, it will not affect the running speed of the template matching. Significantly, the GPT is introduced into the template matching algorithm, ensuring the accuracy of the matching results. Experiments on real-world images demonstrate that our method performs better than the traditional NCC and SAD algorithms in terms of operation efficiency and accuracy. We also compared with other common image matching algorithms, such as SIFT matching, to further demonstrate its applicability and validity. The simplicity and effectiveness of the proposed method make it suitable for real-time implementation on a borehole-wall unfolded image sequence for an image mosaic.

## 3. Overview of the System

The axial view borehole camera system (AVBCS) was equipped with a wide-angle camera at the front of the probe. The probe was positioned inside a pre-drilled borehole to capture the borehole video. Then, the forward-looking borehole video acquired by AVBCS was used to generate a whole cylindrical borehole-wall panorama using our proposed system with image processing technology.

The proposed system was composed of four main modules, i.e., locating the drilled hole center, generating of borehole-wall unfolded images, matching of unfolded images and an image filtering module. The center location module was mainly developed to extract the drilled hole center from the front view borehole-wall image so as to resist geometric distortion. The drilled hole center hole was also used as the image unwrapping center for the subsequent step. In the second step, the effective annular region was unfolded into a rectangle, namely the borehole-wall unfolded image. The third step was to achieve the mosaic of the unfolded image sequence by image matching technology based on the proposed PTSAD algorithm. In addition, we introduced an image filtering module to eliminate

the invalid frames using three parameters and the valid frames were further stitched into an integral cylindrical borehole-wall panorama. The flowchart of the proposed system is provided in Figure 1.

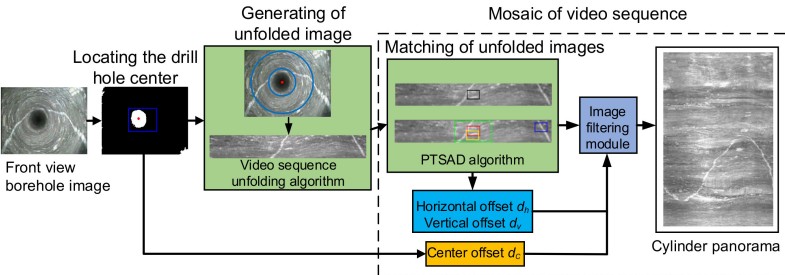

**Figure 1.** Flowchart describing the system overview.

## 4. Methods

### 4.1. Locating Drilled Hole Center

When a camera records videos in a borehole, the probe usually generates a slight vibration and rotation. In this condition, if the geometric center (yellow points in Figure 2) is used for the image unwrapping center, severe geometry distortion would arise in the image unwrapping process. Due to the limited lighting area, light intensity decreases dramatically with the increase of the distance to the probe. Therefore, a visual blind area was formed in the center of the image, as shown in Figure 2. The centroid of the central black area can be represented as the drilled hole center. However, the drilled hole center in each frame (red points in Figure 2) was not fixed. Thus, it was necessary to locate the drilled hole center automatically.

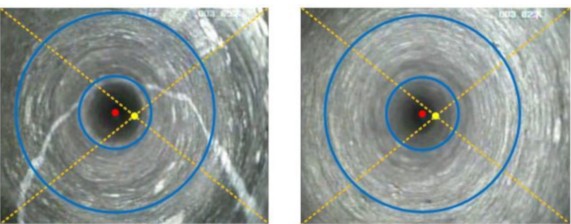

**Figure 2.** The drilled hole center deviation.

According to the characteristics of a front view borehole-wall image, we proposed an automatic location method of the drilled hole center. The architecture of the method is shown in Figure 3. Firstly, gray-scale transformation was used to adjust the gray-scale of the borehole-wall image and accordingly gave prominence to the contrast of the central black area. Secondly, the Otsu algorithm [26] was adopted to segment the image, and then the regions containing the central region were obtained. Thirdly, the region labeling algorithm [27,28] was used to eliminate the small areas and extract different connected regions. Finally, the centroids of all connected regions were counted. Since the drilled hole center was located near the center of the image, the centroids of the non-central regions could be easily excluded by setting an appropriate scope, as shown in the blue box of Figure 3. In this way, we realized the effective localization for the drilled hole center.

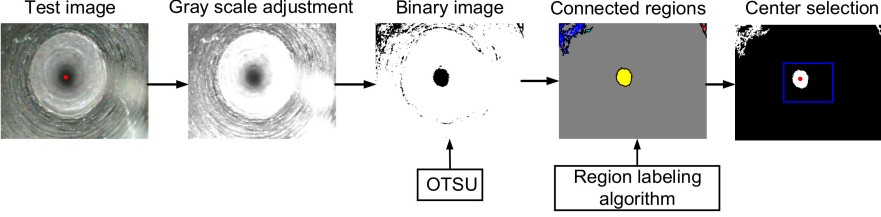

**Figure 3.** Steps involved in locating the drilled hole center.

### 4.2. Generating of Unfolded Image Sequence

After locating the drilled hole center, the annular effective region centered on the drilled hole center could be determined by setting the radii of inner and outer circles. Then it was used to generate the rectangular image. As the drilled hole center of each frame could be located automatically, the radii of the inner and outer circles were set to fixed values, so that each frame had the same annular region. Due to the continuity of video, when the drilled hole centers of two video frames move around, the annular regions will result in the overall displacement. Thus, the mosaic result image will not lose image information by setting a fixed radius, rather it simplifies the video processing.

In order to unfold the forward-looking borehole video into a rectangular image sequence, we proposed a borehole video sequence unfolding method, which designed a virtual coordinate graph (VCG) to assist with coordinate mapping (CM), thereby reducing the calculations of coordinate transformation in the unwrapping process of each video frame and improving the unwrapping efficiency of the video sequence.

The schematic diagram of the image unwrapping process is presented in Figure 4. The relationship between the polar and rectangular coordinates was established to map pixels in the annular effective region to the unfolded image. Subsequently, the pixels of the annular region were used to fill the unfolded image by the reverse-mapping method [29], as described below.

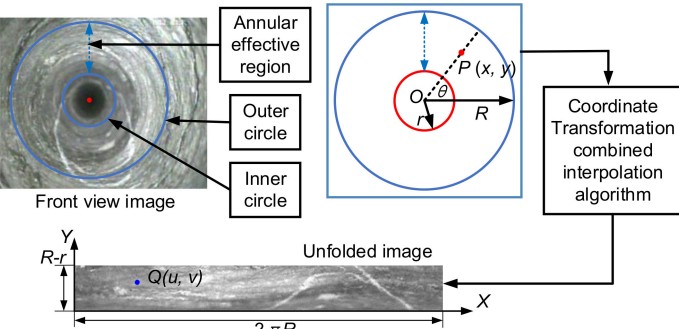

**Figure 4.** Unwrapping process of front view borehole-wall image.

In the front view image, $O\ (x_0, y_0)$ is the drilled hole center, $r$ the radius of inner circle and $R$ is the outer circle. $P\ (x, y)$ is a point in the annular region and its corresponding point in the unfolded image is $Q\ (u, v)$. The unfolded image is constructed in advance and the coordinate of point $Q$ is used to calculate the coordinate of point $P$. Then the pixel value of point $P$ is assigned to point $Q$. The relationship between two points in two spaces is as follows:

$$\begin{cases} x = x_0 + (r + v) \sin \theta \\ y = y_0 + (r + v) \cos \theta \quad , \\ \qquad \theta = \frac{u}{R} \end{cases} \tag{1}$$

where, $\theta$ is the polar angle of the annular image, $(x_0, y_0)$ is the coordinate of the center, $(x, y)$ is the pixel coordinate of point $P$ and $(u, v)$ is the pixel coordinate of point $Q$. The length of the unfolded image is $2\pi R$ (with the circumference of the largest radius), and the width is $R-r$ (effective width for the analogy ring). In this research, the size of the front view image was 300 pixels × 238 pixels. To ensure that the unfolded image could include more effective information, the size of $r$ and $R$ were set to 50 and 110 respectively. Thus, the size of the unfolded image was 660 pixels × 60 pixels ($M = 2\pi R = 660$, $N = R-r = 60$).

The front view borehole-wall image captured by AVBCS is a kind of distorted image with its imaging characteristics being that the pixels on the ring of the same radius are uniformly distributed, but they are not uniform in the radial direction. Therefore, image quality of the inner ring degrades more than its outer ring. To generate the unfolded images, we used the interpolation algorithm to fill the

missing pixels for the inner rings. The nearest neighbor, bilinear and cubic convolution interpolations are three major interpolation algorithms used in image processing [30]. Nearest neighbor interpolation has the advantages of less operations and faster speed, while the unfolded image has low quality, and the problems of contour jaggies and mosaic appearance are serious. The application of bilinear interpolation can obtain the satisfactory result, but the computation is larger than the nearest neighbor interpolation. Although the calculation precision of cubic convolution interpolation is better than of the other two methods, this involves higher matrix computing, and thus the quantity computed and time expended will increase obviously [31]. In this paper, to achieve the real-time and high precision, the bilinear interpolation algorithm was applied in the image unwrapping process.

In the process of generating the rectangular image sequence from the borehole video, the mathematical model of the unwrapping algorithm was exactly the same, thus the calculation of coordinate conversion for each video frame was a repetitive task. In this paper, we designed a virtual coordinate graph (VCG) to reduce the computational complexity.

Using the combination of the VCG and CM to unfold the borehole video sequence is shown in Figure 5. Firstly, we needed to pre-compute the relationship of coordinates between the rectangular image and the front view image, namely $(u, v) \rightarrow (x, y)$. The relationship $(u, v) \rightarrow (x, y)$ was used to build the VCG, which stored a one-to-one mapping of the coordinate between the point $Q$ and point $P$. Then, we used the VCG to directly invoke $(u, v) \rightarrow (x, y)$ when unfolding the next frame, and the pixel value of point $P$ was calculated with four adjacent pixels $P_1$, $P_2$, $P_3$ and $P_4$ in the front view image by the bilinear interpolation algorithm. After that, the pixel value of point $P$ was assigned to the point $Q$ in the rectangular image, namely $P(x, y) \rightarrow Q(u, v)$. This process needs no computation of a transformation matrix between the Cartesian coordinates and polar coordinates, and the processing speed of the video sequence is greatly improved.

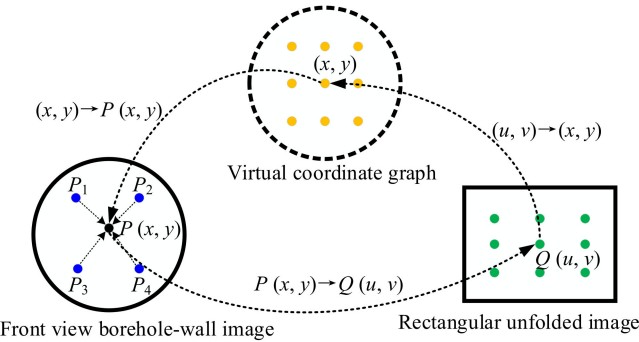

**Figure 5.** Virtual coordinate graph.

### 4.3. Mosaic of Borehole Video

#### 4.3.1. Mosaic Principle of the Borehole Wall Image

After locating the drilled hole center, the movement of the camera in the borehole could be decomposed into the axial beeline motion and circumferential rotation motion. In Figure 6, the frames A and B were used to simulate two continuous frames of front view borehole-wall images, where the green region was the annular effective region used for the image unwrapping and the blue oval was the fracture. While shooting the frame B, the camera produced a forward motion and rotation motion along the central axis of the drilled hole, which corresponded to the radial offset and angular offset $(\beta - \alpha)$ relative to frame A, respectively.

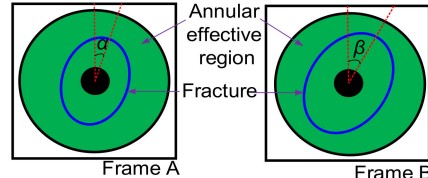

**Figure 6.** The relative motion of two front view borehole-wall images.

The corresponding two rectangular images were created from two continuous frames of front view images by the proposed image unwrapping algorithm, and then were used for the image mosaic, as depicted in Figure 7. The relative displacement of rectangular images could be divided into vertical and horizontal offsets, corresponding to the radial and angular offsets of front view images respectively. These two offset values constituted the final mosaic parameters and were used to implement the mosaic of the unfolded image sequence in the axial direction and rotation correction in the horizontal direction, respectively. Since the rectangular images were generated from the annular images, the horizontal displacement region caused by camera rotation (the yellow rectangular area shown in Figure 7) could be cut to the red blank dotted rectangular area to realize the rotation correction. Therefore, the rectangular images were stitched together as one single cylindrical panorama in the axial direction according to the vertical offset between consecutive frames. In the image mosaic process, template matching (TM) was chosen to calculate the mosaic parameters.

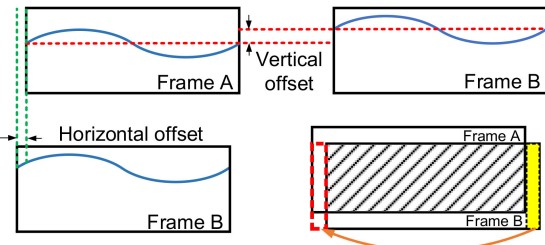

**Figure 7.** The mosaic principle of unfolded images.

The objective of TM is to establish the correspondence between the source image and the examined image. TM is a process to locate the position of predefined template in the examined image, which is widely used in object detection, target tracking, image mosaic and so on. There are many kinds of TM methods, the application of which should depend on the specific circumstances of provided images and intended application. TM is illustrated in Figure 8. A template is chosen in a source image, and a search area (green area in Figure 8) is defined in an examined image. In the matching process, to find the sub-image matching with the template, the template is shifted through every location of the search area in the examined image to recognize the most similar candidate image of the same size (image part under template). In this paper, the first frame of the video sequence will initially act as the source image, and the second frame was used as the examined image. These two images were stitched together into a whole image as a new source image by the template matching method, and then matched with the third frame to update the source image. In this way, the mosaic of the whole video sequence could be successfully completed.

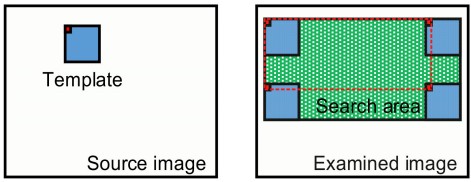

**Figure 8.** Schematic diagram of the template matching.

### 4.3.2. Proposed PTSAD Algorithm

The accuracy and operational efficiency are two important factors when choosing the TM algorithm. The method that gives more accurate results is often associated with more computational expense and procedure complexity, whereas the faster method is simpler but leads to an inaccurate result. To ensure the accuracy of the mosaic for each video frame without losing operational efficiency and while meeting the real-time application requirement of video image processing, we selected an appropriate template and a search area, and proposed the PTSAD method to measure the similarity between the template image and target image.

- Template selection

The best template in the source image includes as many distinctive features as possible. However, selecting a larger template requires more computations for the image matching. On the other hand, a smaller template requires less computation, but few distinctive features will reduce the matching performance.

In this paper, because the resolution of the front view image decreased gradually in the radial direction, the resolution of the borehole-wall unfolded image was also reduced from top to bottom. Accordingly, the template was selected at the center of the image, as shown by the red box in Figure 9.

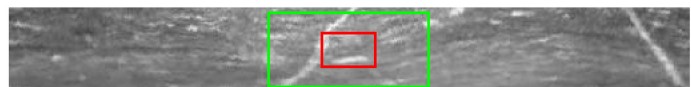

**Figure 9.** The selection of the template and search area.

- Search area selection

There are two ways to reduce the running time of the template matching algorithm: One is to decrease the cost of computing in each matching point, and the other is to narrow the search area. In the traditional matching method, such as full-search template matching (FSTM) [32], a large number of search areas are invalid, causing a heavy computational cost. In order to reduce the unnecessary computational time of the template matching in whole video sequences, an effective measure reducing the search area is also very significant.

Considering that the displacement between two adjacent video frames was relatively small, the search area could be controlled within a proper range around the template, as shown by the green box in Figure 9, where the computational cost could be reduced dramatically by minimizing the search range. The search area selection in this paper eliminated a great number of unnecessary points, and the operation speed was tripled, at least.

- Similarity measure method of PTSAD

As the similarity measure method, the NCC (normalized cross correlation) and SAD (sum of absolute difference) algorithms are the most popular and widely applied to the TM [22]. NCC is more robust against illumination changes than SAD, nevertheless, NCC is more time-consuming than SAD. The SAD algorithm is typically used for the template matching of the video image due to its simplicity and higher operating speed. In this paper, to improve the accuracy of the SAD algorithm, a gray projection transformation (GPT) was introduced into the TM method, which provided a new PTSAD algorithm. The architecture of the PTSAD algorithm is shown in Figure 10.

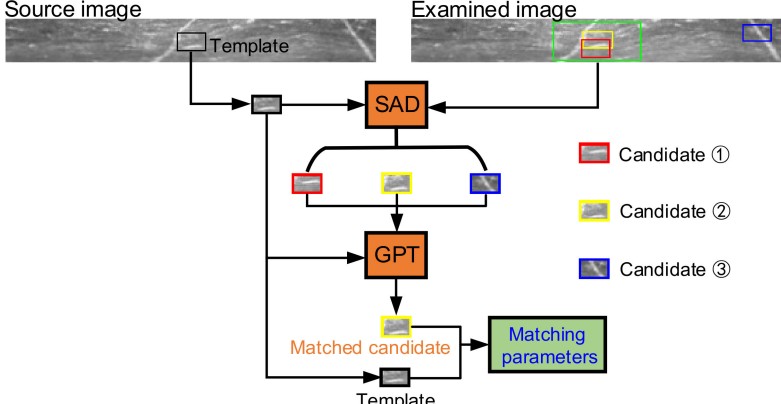

**Figure 10.** The architecture of the proposed projection transformation sum of the absolute difference (PTSAD) algorithm.

SAD measures the degree of similarity between a template and the examined image. It is a method that sums the total differences of the gray level of each pixel between the template and the examined image. Suppose $T$ is the template of size $m \times n$ and $E$ is the examined image of size $u \times v$, where $u > m$ and $v > n$. SAD at point $(x, y)$ in the examined image is defined as follows [33,34]:

$$SAD(x,y) = \sum_{i=0}^{m-1} \sum_{j=0}^{n-1} \left| E(x+i, y+i) - T(i,j) \right|,$$
(2)

where, $E(i,j)$ and $T(i,j)$ are the pixel values at coordinate $(i,j)$ in the examined image and template respectively. $(x, y)$ represents a pixel position of the current search point. The smaller the value of SAD, the more similar to the template, and the candidate image with the minimum SAD value is usually regarded as the best matching image.

The GPT reduces a two-dimensional image to a one-dimensional gray scale projection curve. It can detect the gray-scale changes of the image and significantly simplifies calculation. Then, by computing the correlation coefficients of 1D vectors and comparing the correlation between the template and different candidate images, the sub-image matching with the template can be recognized from candidate images. The GPT is a process in which the pixel gray value is superimposed in rows or columns by Equations (3) and (4) respectively, as follows [35]:

$$R(i) = \frac{1}{n} \sum_{j=1}^{n} T(i,j),$$
(3)

$$C(j) = \frac{1}{m} \sum_{i=1}^{m} T(i,j),$$
(4)

where, $T(i,j)$ is the gray value of point $(i,j)$, $R(i)$ is the sum of gray values in row $i$ and $C(j)$ is the sum of gray values in column $j$. The gray scale projection curve in the x- or y-direction reflects the gray distribution features of image. Thus, this property can be used to identify different candidate images.

Suppose $T(x, y)$ is the template and $C_i(x, y)$ are the candidate images, where $i = 1, 2$ and 3. We projected $T(x, y)$ and $C_i(x, y)$ to the *X*-axis and obtained 1D vectors $T$ and $C_i$, respectively. The correlation coefficient of the $T$ and $C_i$ was calculated by Equation (5):

$$R_i(x,y) = \frac{E(T \cdot C_i) - E(T) \times E(C_i)}{\sqrt{D(T) \times D(C_i)}}, i = 1,2,3,$$
(5)

where, $E(T)$ and $E(C_i)$ are the mean values of vectors $T$ and $C_i$, respectively. $D(T)$ and $D(C_i)$ are the variance of vectors $T$ and $C_i$, respectively.

The correlation coefficient of vectors in the *Y*-axis can be obtained in a similar way. The max of $R_i$ $(x, y)$ was used to find the candidate image matching with the template from three candidate images. Subsequently, the matching position was acquired accurately and then the matching parameters between two images were obtained.

In summary, the flow of the proposed PTSAD algorithm was as follows:

Step 1: Load two continuous borehole-wall unfolded images as the source image and examined image respectively, and select the template in the source image.

Step 2: Move the template over the preset search area in the examined image and simultaneously calculate the SAD value of the template and the candidate images under the template, and store them in an array.

Step 3: Extract the candidate images of three minimum SAD values in the array (the SAD value from small to large in sequence of colors: Red, yellow and blue).

Step 4: Calculate the 1D projection vectors of the template and the three candidate images in both the X-axis and Y-axis by GPT, then compute the correlation coefficients of these vectors and find the candidate image, which has the maximum correlation coefficient, that is the candidate image matching with the template.

To sum up, the pseudo-code of image matching based on PTSAD is summarized in the Algorithm 1, as following.

---

**Algorithm 1** Pseudo-code: Projection Transformation SAD (PTSAD)

---

**Input:** Source image $P_{m \times n}$, examined image $Q_{m \times n}$, where $m \times n$ is the size of image.
**Output:** Horizontal offset $d_h$, vertical offset $d_v$.

1.   **Set** $T_{u \times v}(x, y) = P_{u \times v}(x, y)$, $P^{u \times v} \in P^{m \times n}$, where $u \times v$ is the size of image, Search area $S_a \in R^{m \times n}$
2.   **Obtain:** Candidate images set $\{C(1), C(2), \ldots, C(k)\}$, with step size 1 pixel
3.   **Compute:** SAD value of $T_{u \times v}(x, y)$ and $\{C(1), C(2), \ldots, C(k)\}$
4.   **for** $s = 1, 2, \ldots, k$
5.   $C_{SAD}(s) = \sum\limits_{i=0}^{U-1} \sum\limits_{j=0}^{V-1} \left| C_s(x+i, y+i) - T(i, j) \right|$
6.   **end for**
7.   **Obtain:** SAD value of candidate images $\{C_{SAD}(1), C_{SAD}(2), \ldots, C_{SAD}(k)\}$
8.   Candidate images of three minimum SAD value: $C(R)$, $C(G)$, $C(B)$, where $C(R) < C(G) < C(B)$
9.   **Compute:** Projection vectors in X-axis of $T(x, y)$, $C(R)$, $C(G)$ and $C(B)$
10.   **for** $i = 1, 2, \ldots, u$ **do**
11.   **for** $j = 1, 2, \ldots, v$
12.   $T_h(i) = \frac{1}{v} \sum_1^v T(i, j)$; $C_h(i) = \frac{1}{v} \sum\limits_1^v C(i, j)$
13.   **end for**
14.   **end for**
15.   **Obtain:** $T_h$, $C_h(R)$, $C_h(G)$ and $C_h(B)$
16.   **Compute:** Correlation coefficient
17.   $R_i(x, y) = \frac{E(T \cdot C_h(i)) - E(T) \times E(C_h(i))}{\sqrt{D(T) \times D(C_h(i))}}, i = R, G, B$
18.   **Obtain:** In X-axis: $R_h(R)$, $R_h(G)$, $R_h(B)$, and In Y-axis: $R_v(R)$, $R_v(G)$, $R_v(B)$
19.   **Compute:** Linear constant $Z$
20.   $Z = \alpha R_h(x, y) + R_v(x, y)$
21.   **Obtain:** $Z(R)$, $Z(G)$, $Z(B)$, where $Z(B) > Z(R) > Z(G)$
22.   **Compute:** Horizontal offset $d_h$, vertical offset $d_v$
    $d_h = |x - u|$, $d_v = |y - v|$, where $(x, y)$ and $(u, v)$ are the center coordinates of $T(x, y)$ and $C(B)$

---

### 4.4. Image Filtering Module

Ideally, the probe goes forward along the central axis of the pre-drilled borehole with uniform motion and without any shaking. However now this process works in manual mode, it will affect the motion of the probe directly. The irregular movement of the probe usually involves vibration, rotation and stagnation. Vibration and rotation will cause deviation between the geometric center of the image and the drilled hole center, which can lead to serious image geometric distortion during the image unwrapping process. Stagnation produces a large number of duplicate images. In general, the vibration and rotation of a probe are always accompanied with the stagnation of probe. The impact of slight vibration and stagnation can be resolved by the center location module (Section 4.1) and horizontal offset correction of unfolded images (Section 4.3), respectively. However, the stagnation of a probe, and severe vibration and stagnation are very hard to settle by image processing techniques. Therefore, the invalid video frames caused by these factors should be identified and removed before generating a cylindrical panorama.

After locating the drilled hole center, the offset between the geometric center and the drilled hole center can be computed, which is represented by $d_c$. The horizontal and vertical offset can be obtained from the matching parameters, which are represented by $d_h$ and $d_v$ respectively. The parameters $d_c$, $d_h$ and $d_v$ are used to monitor the vibration, rotation and stagnation respectively. Before generating the cylindrical panorama, if the value of one of the three parameters value exceeds the normal range value, the frame is regarded as an invalid frame and cannot be used in the image mosaic process. The threshold range for $d_c$, $d_h$ and $d_v$ is preset by the following.

$$\begin{cases} 0 \le d_c \le 7 \\ 0 \le d_h \le 3 \\ \quad d_v > 2 \end{cases}. \tag{6}$$

## 5. Experimental Results and Analysis

### 5.1. Image Acquisition System

In this paper, the YTJ20 type of axial view borehole camera system (AVBCS) was selected, which was composed of a camera, video transmission line, digital depth counter and control box. (as illustrated in Figure 11). The diameter and length of the probe was 25 mm and 100 mm respectively. The equipment had 8 h of continuous running time and 2 GB of storage capacity.

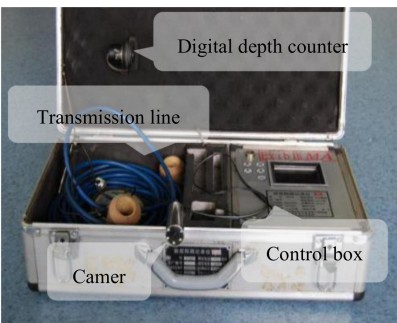

**Figure 11.** YTJ20 type of an axial view borehole camera system (AVBCS).

In order to observe the geological structure, engineers position the probe in a pre-drilled borehole to capture videos. A camera was mounted on the top of the probe, which could obtain videos of the effective lighting area, as illustrated in Figure 12. The resolution of video sequences obtained by the AVBCS was about 0.1 mm and the acquired images were both of 320 pixels × 240 pixels.

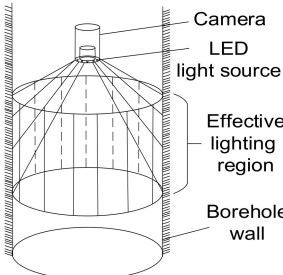

**Figure 12.** Schematic drawing of an axial view camera.

During the detection process of the drilling hole, the AVBCS captured borehole videos and sent them to our data-processing system. The front view panoramic borehole-wall image, which is a mapping of the three-dimensional borehole-wall image within the effective lighting area to the two-dimensional image, was obtained from the borehole video. Figure 13a,b shows the front view images of four successive frames obtained from two borehole videos, respectively. The boreholes presented in this paper belonged to the field of exploration in a coal mine, which were drilled in the roof strata of a ventilation roadway. The stratum of a deep shaft coal roadway is mainly composed of sandstone, siltstone, mudstone and coal seam. The depth of boreholes was about 5.7 to 5.8 m.

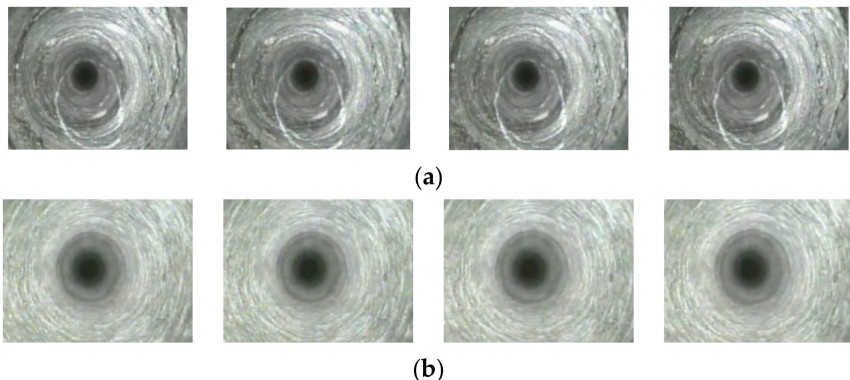

**Figure 13.** Front view borehole-wall images. (**a**) Sample sequence of video 1; (**b**) sample sequence of video 2.

### 5.2. Center Positioning and Image Unwrapping Experiments

We first evaluated the proposed center positioning algorithm on different types of front view borehole-wall images. The images were obtained from forward-looking borehole videos taken by AVBCS. Figure 14a–c shows the process of center positioning for border images, fracture images and intact rock mass images respectively. From Figure 14, the region containing the drilled hole center was successfully segmented and then the center was extracted by our method accurately. The center of the preset blue box was the geometric center of image and its size was $100 \times 80$ pixels.

Table 2 shows the coordinates of the drilled hole center for Figure 14. The parameter $d_c$ is the distance between the geometric center of the image and the drilled hole center. The center coordinate of the drilled hole was different in each frame and the offset varied in a certain range. In addition, 500 continuous borehole-wall images were used to evaluate our algorithm in locating the drilled hole center and the results show that the proposed method was feasible and effective for the central location of various structural borehole-wall images.

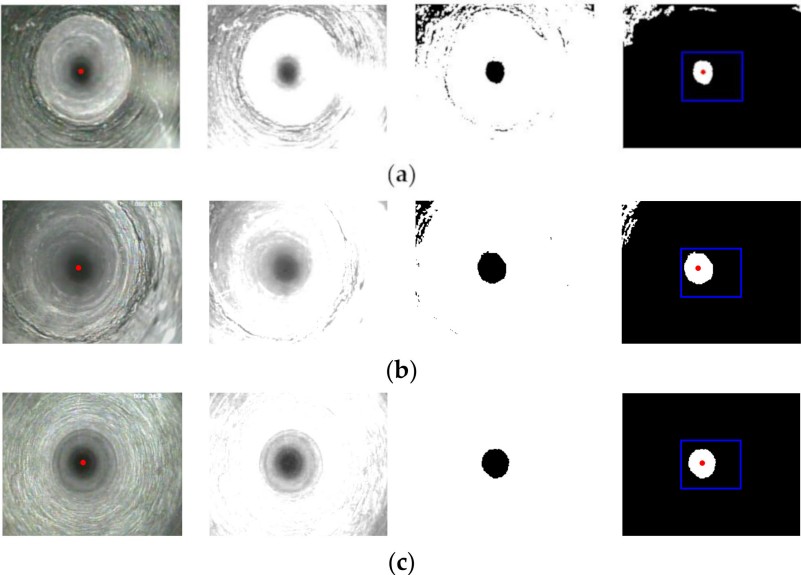

**Figure 14.** The center location for three types of borehole-wall images. (**a**) Process of center positioning for border image; (**b**) process of center positioning for fracture image and (**c**) process of center positioning for intact rock mass image.

**Table 2.** Drilled hole center coordinates.

| Image | X-Coordinate | Y-Coordinate | $d_c$ |
|---|---|---|---|
| Figure 14a | 142 | 111 | 3 |
| Figure 14b | 143 | 109 | 3 |
| Figure 14c | 148 | 109 | 3 |

For the front view borehole-wall image, we set the radii of the inner circle and outer circle to 50 and 110, respectively. The size of the rectangular image generated from the annular region was the 660 pixels × 60 pixels ($M \times N$, $M = 2\pi R = 660$, $N = R - r = 60$). Figure 15b,c shows the borehole-wall unfolded images generated by the bilinear and cubic convolution interpolations, respectively. The image effect of bilinear interpolation was very similar to that of the cubic convolution interpolation. Thus, it would not affect the overall image mosaic effect.

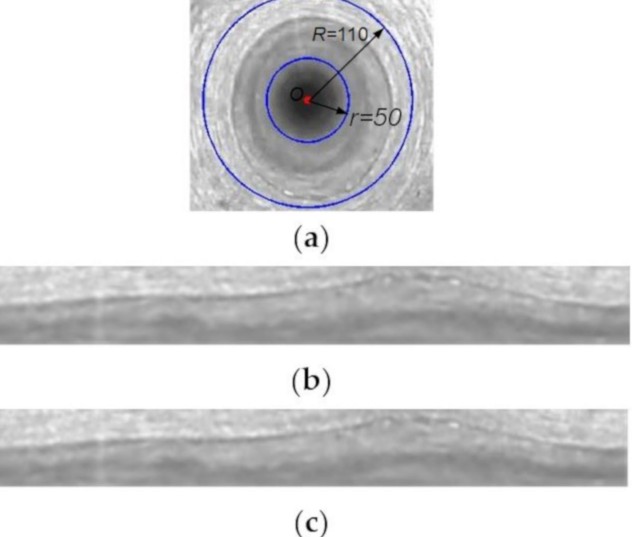

**Figure 15.** Borehole unfolded images. (**a**) Front view borehole-wall image; (**b**) unfolded image obtained by the bilinear interpolation and (**c**) unfolded image obtained by the cubic convolution interpolation.

*5.3. Borehole Wall Unfolded Images Matching Experiment*

Image matching is the key step to stitch a complete cylindrical panorama from a borehole video. In this paper, we proposed a PTSAD algorithm to realize the matching of the video image sequences. The matching effect had influenced the accuracy and efficiency of generating the cylindrical image directly. In this section, two adjacent unfolded images, as shown in Figure 16a,c, were used to confirm the validity of the proposed PTSAD algorithm for an automatic image mosaic. In order to compare the feasibility and effectiveness of the proposed method with other established algorithms, we conducted the same experiments on the image matching by the feature-based of SIFT [36] and the full-search (FS) of NCC and SAD [37].

Figure 16a,c are the resource image and the examined image in the template matching respectively. Figure 16b is the template extracted from the resource image, that is, the black rectangle in Figure 16a. The template shifted over the whole area in the examined image and compared to the candidate images of the same size as the template. In Figure 16c, red, yellow and blue rectangles are the three candidate images with the minimum SAD value in ascending order. It is obvious that the candidate image of the yellow rectangle was the most similar to the template, whereas the minimum SAD value was the candidate image of the red rectangle. Thus, the search result could sometimes be inefficient when drawn from the traditional SAD algorithm.

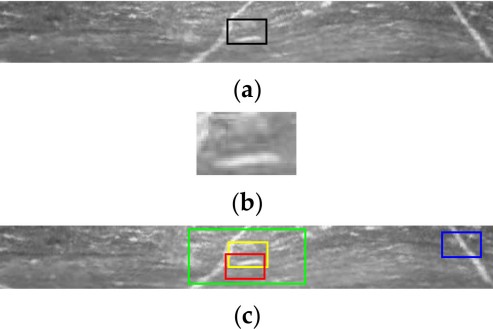

(a)

(b)

(c)

**Figure 16.** Three candidate images in fracture borehole image. (**a**) Resource image; (**b**) template and (**c**) examined image.

Similarly, when the borehole-wall images had smooth texture or no distinctive features, as shown in Figure 17, the candidate image of the minimum SAD value was not the candidate image matching with the template. In Figure 17c, the candidate image of the third-smallest SAD value (blue rectangle) was the candidate image matching with the template.

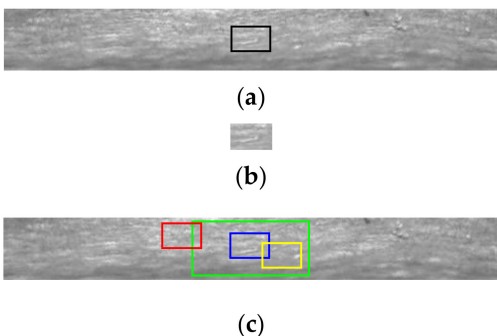

(a)

(b)

(c)

**Figure 17.** Three candidate images in a smooth borehole image. (**a**) Resource image; (**b**) template and (**c**) examined image.

In order to identify the best candidate image from three candidate images, the GPT was used as the decision module in our PTSAD algorithm. By computing the correlation coefficients of one-dimensional projection vectors obtained by GPT and comparing the correlation between the template and the three

candidate images, the candidate image matching with the template could be recognized from the candidate images.

Figure 18a is the template image, namely, the black rectangle in Figure 16a. Figure 18(a1,a2) are horizontal and vertical projection curves of the template respectively. Figure 18b–d are the three candidate images, namely the yellow, red and blue rectangles shown in Figure 16c. Their corresponding projection curves are shown in Figure 18. The results of the projection curves show that the candidate image matching with the template (Figure 18(b1,b2)) had a more similar waveform compared to the template (Figure 18(a1,a2)) than the other two candidate images.

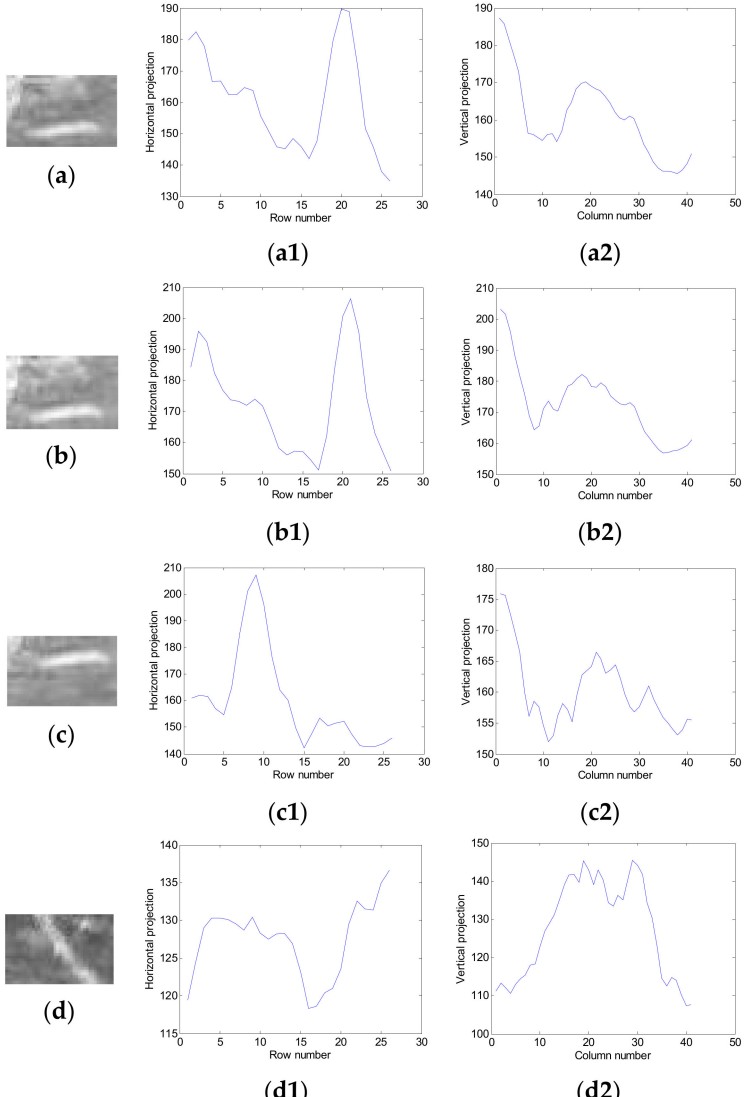

**Figure 18.** The horizontal and vertical projection curves of the template and candidate images. (**a**) Template; (**a1**) horizontal projection curve of the template and (**a2**) vertical projection curve of the template. (**b**) Yellow candidate image; (**b1**) horizontal projection curve of the yellow candidate image and (**b2**) vertical projection curve of the yellow candidate image. (**c**) Red candidate image; (**c1**) horizontal projection curve of the red candidate image and (**c2**) vertical projection curve of the red candidate image. (**d**) Blue candidate image; (**d1**) horizontal projection curve of the blue candidate image and (**d2**) vertical projection curve of the blue candidate image.

The correlation coefficient of the projection vector was calculated by Equation (5). As shown in Figure 19, TY, TR and TB were the correlation coefficients between the template and the yellow, red and blue candidate images respectively. HCC and VCC are the correlation coefficients of the horizontal

projection vector and vertical projection vector respectively. Consistent with the results shown in Figure 18, the yellow candidate image (Figure 18b) was the most similar candidate to the template and had the maximum correlation coefficients in the horizontal projection and vertical projection. Thus, the sum of the correlation coefficients for the horizontal projection vector and vertical projection vector could be used to distinguish the candidate image matching with the template from other candidate images.

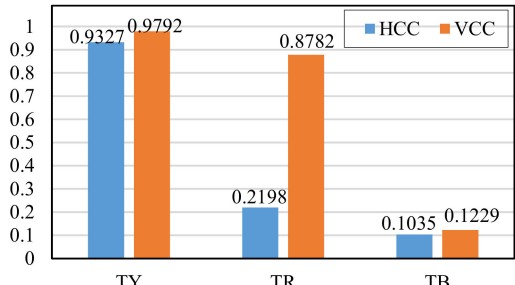

**Figure 19.** Correlation coefficients of the three candidate images.

The quantity of candidate images that are input to the GPT module can affect the speed and accuracy of template matching. We calculated the matching accuracy of a different number of candidate images for testing 60 continuous borehole-wall unfolded images. The result shows that matching accuracy achieved the best results when the number of candidate images exceeded three. Therefore, to reduce the matching time, we extracted three candidate images with minimum SAD values to be input to the GPT module.

The feature-based SIFT is another commonly used method in template matching, thus we conducted the same experiments with SIFT template matching. The results of SIFT matching are shown in Figure 20. Among these, Figure 20(a1,b1) are two successive frames of a borehole-wall unfolded image, which contain more texture features. Therefore, more feature points could also be detected and then elaborated by image descriptors in both images, as shown in Figure 20(a1,b1). Given enough feature points of potent matches, it was possible to achieve matching between the two images as shown in Figure 20(c1). The feature-based methods depend on supposing that enough effective feature points can be detected and paired successfully. If there were no clear features in the images, these methods would have a high matching error rate. Due to the decrease of effective feature points (in Figure 20(a2,b2)), there were a certain amount of mismatching points in the SIFT matching result, as shown in Figure 20(c2).

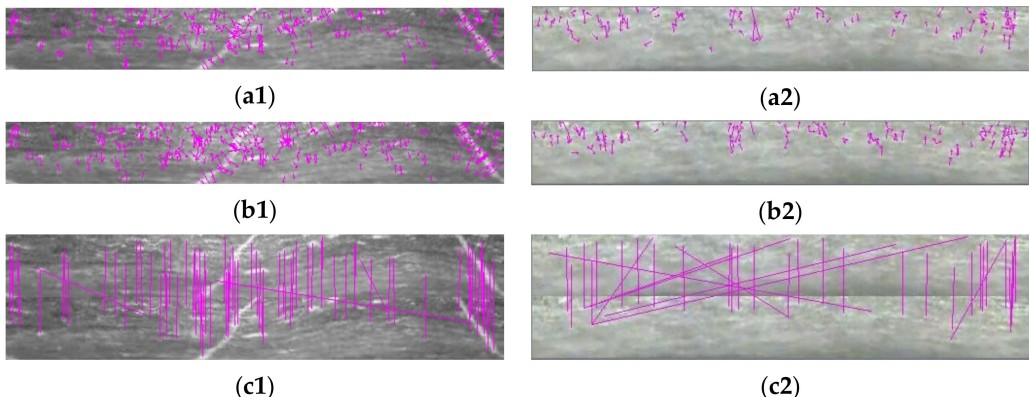

**Figure 20.** The result of SIFT matching. (**a1**) and (**b1**) are SIFT extraction results of two successive frames of fracture images, (**c1**) is the matching result; (**a2**) and (**b2**) are SIFT extraction results of two successive frames of intact rock mass images and (**c2**) is the matching result.

Finally, traditional template matching algorithms such as NCC and SAD, were also compared. An experiment was performed on the same two borehole-wall unfolded images to analyze the accuracy and efficiency of different template matching algorithms. A template of size $25 \times 40$ pixels (Figure 21b) was extracted from the source image of size $660 \times 60$ pixels (Figure 21a). Then the template was used in the examined image (Figure 21c) for the template matching process.

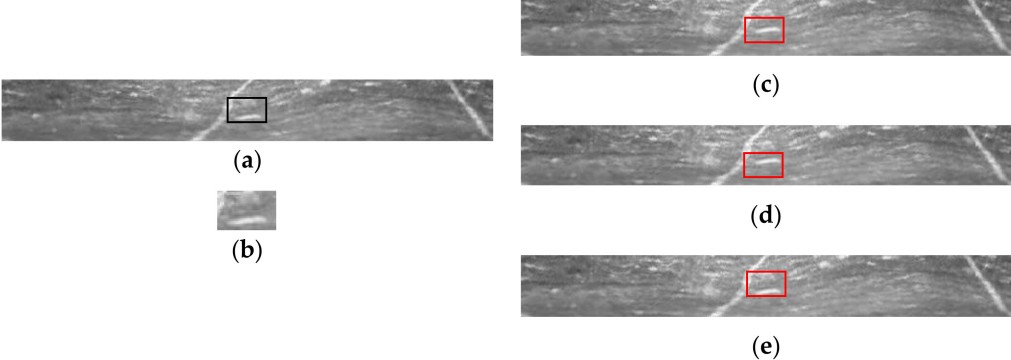

**Figure 21.** The matching result of two consecutive frames. (**a**) Source image; (**b**) template; (**c**) matching result of normalized cross correlation (NCC); (**d**) matching result of SAD and (**e**) matching result of the proposed PTSAD method.

Figure 21c–e shows the matching result of full-search NCC, full-search SAD and the proposed PTSAD methods, respectively. The experimental results indicate that the NCC and SAD algorithms could not locate the template accurately because of the interference of several factors such as image rotation, image blur, scale variation and illumination change between the examined image and source image. Apparently, it is obvious that the proposed method performed better than NCC or SAD, as shown in Figure 21e.

### 5.4. Mosaic Results of Cylindrical Panorama

In this experiment, two videos obtained by AVBCS, recorded in a practical mining engineering application, were provided to verify the performance of our system. These two videos had 225 frames and 571 frames respectively.

The first video sequence was inputted into the image filtering module to identify the invalid frames. The center offset $d_c$, horizontal offset $d_h$ and vertical offset $d_v$ for 225 frames were acquired and are shown in Figure 22. In Figure 22, the values of $d_c$ and $d_v$ for most frames were limited to a small range, indicating that the vibration and rotation of the probe were generally minimal. However, for frame 26 to frame 51, these two values exceeded the permitted value. Meanwhile, the values of $d_h$ for these frames were always 0 or 1. This means that the probe had stopped, accompanying strong vibration and rotation. Therefore, these frames should be considered as invalid frames.

After eliminating the invalid frames, the two videos had 200 frames and 540 frames, respectively. Subsequently, these two video sequences were used to generate two complete borehole-wall cylindrical panoramas by our method.

The execution time and the false matching rate for these two video sequences by different matching methods are provided in Tables 3 and 4, respectively. They show that the proposed method offered lower computational cost as compared to most other algorithms. The proposed method was nearly five times faster than the full-search NCC and three times faster than the full-search SAD. There are many different types of borehole-wall images in these two videos sequences such as border images, fracture images and intact rock mass images. Most significantly, experiments show that the proposed matching method could fulfill the matching tasks accurately for each different texture of the borehole-wall image. The mosaic results of these two video sequences are shown in Figures 23 and 24, respectively.

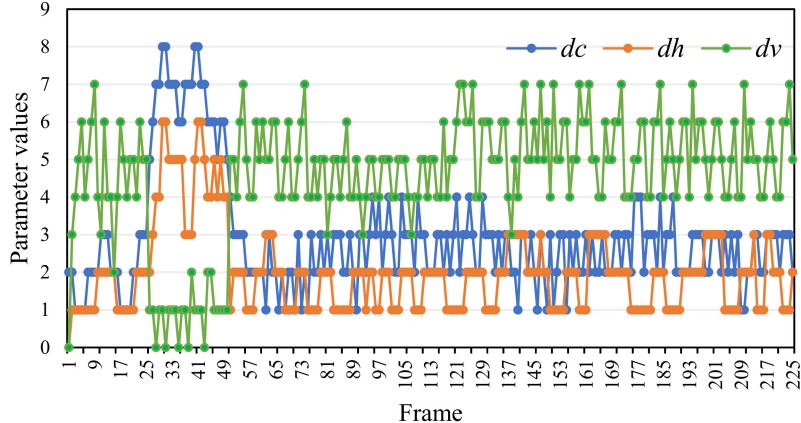

**Figure 22.** The value of $d_c$, $d_h$ and $d_v$ for 225 frames in the first video.

**Table 3.** Execution time and false matching rate of different matching methods for the first video.

| Method | Exec. Time (s) | False Matching Rate (%) |
|---|---|---|
| SIFT | 303 | 19 |
| Full-search NCC | 267 | 45 |
| Full-search SAD | 154 | 59 |
| Proposed PTSAD | 53 | 0 |

**Table 4.** Execution time and false matching rate of different matching methods for the second video.

| Method | Exec. Time (s) | False Matching Rate (%) |
|---|---|---|
| SIFT | 802 | 23 |
| Full-search NCC | 698 | 39 |
| Full-search SAD | 413 | 46 |
| Proposed PTSAD | 134 | 0 |

Comparing the mosaic results and considering the matching results and execution time, we derived several advantages of the proposed PTSAD algorithm:

1. On the matching performance, considering the characteristics of the borehole-wall image sequences, the PTSAD algorithm ensures the accuracy and reliability of image matching by introducing the GPT process to the SAD. It has solved the problem of the change between the template and examined image.
2. On the computing time, the PTSAD algorithm takes the optimal search area and selects the simpler SAD algorithm. Thus, its computational speed is much faster than the SIFT and full-search NCC and SAD algorithms.

From the intercepted front view images from borehole videos, as shown in Figures 23 and 24 on the left, we could only qualitatively observe the features of coal rock structure and the interface between rock and orebody, but have an obvious weakness in studying the developing degree of the fractural structure plane or the distribution and thickness of the ore vein [38,39]. However, all these problems could be solved by using the borehole-wall cylindrical panoramas as shown in Figures 23 and 24 on the right, where both the integral and detailed information of a borehole such as the fracture distribution, rock mechanical properties and seam thickness can be obtained [40,41].

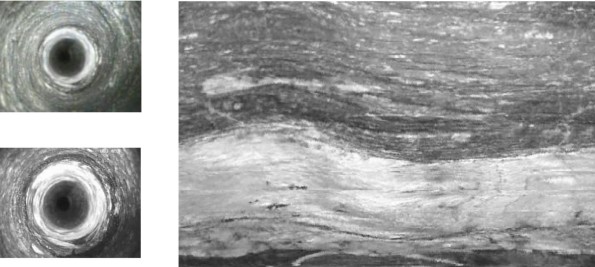

**Figure 23.** Borehole-wall cylindrical panorama generated from the first video.

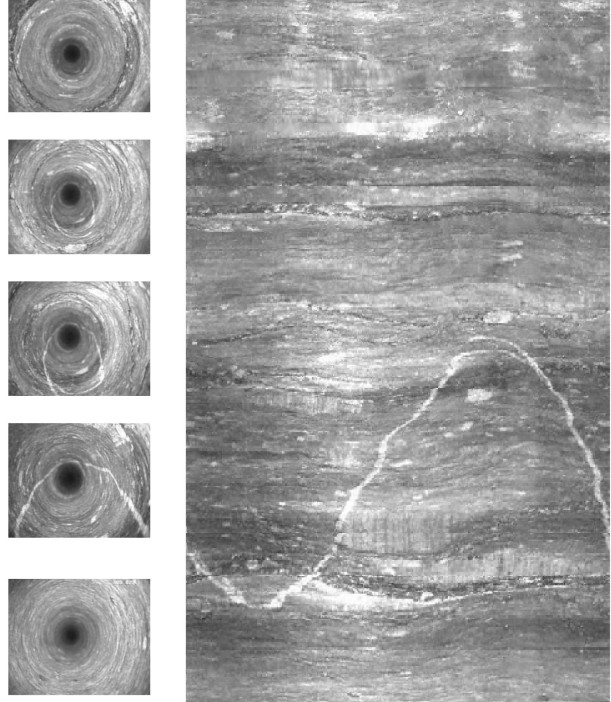

**Figure 24.** Borehole-wall cylindrical panorama generated from the second video.

## 6. Conclusions

This paper proposed a novel approach to generate a complete borehole-wall cylindrical panorama from the forward-looking borehole video acquired by the axial view borehole camera system (AVBCS). Specifically, our system consisted of three main modules: Drilled hole center positioning, front view image unwrapping and an unfolded image mosaic. Through the image unwrapping module, the annular effective regions in front view image sequences were unfolded to the rectangular unfolded image sequences. Thus, it could convert the forward motion to the translational motion in respect of the motion vector estimation. Then, in the image mosaic process, we presented a new PTSAD algorithm to achieve video image matching, which introduced the gray projection transformation (GPT) into the sum of absolute difference (SAD) algorithm to ensure the accuracy of the matching results. In addition, we proposed drilled hole center positioning and image filtering modules to eliminate the impact of camera vibration and remove the invalid frames respectively. The experiments verified that our system could generate the borehole-wall panorama effectively and precisely.

With our method, the forward-looking borehole video could be converted to the corresponding borehole-wall cylindrical panorama. Then engineers could analyze the borehole information quantitatively using the cylindrical panorama, including features such as the distribution and width of a fracture, the thickness and inclination of a coal seam, the strike, dip, dipping direction and developing degree of the structure plane. Furthermore, the proposed method provides a new idea for the video

image mosaic of the forward movement camera. However, complexities involving filling materials in fracture space, studying the lengths, width and inclination of different fractures was not accomplished in this study.

In our future work, we will consider studying the geometric parameters of rock mass structural planes and extending our work to multiple fields, such as pipeline inspection, nondestructive evaluation of gun muzzles and image analysis of an endoscopic video in medical applications.

**Author Contributions:** Z.D. conceived the ideas, designed the experiments, and wrote the paper. Z.D. and M.C. performed the experiments and analyzed the results. Y.G. and L.R. revised the manuscript.

**Funding:** This research received no external funding.

**Conflicts of Interest:** The authors declare no conflict of interest.

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
