# Peer review of "Generating a Cylindrical Panorama from a Forward-Looking Borehole Video for Borehole Condition Analysis"

_applsci, doi:10.3390/app9163437_

Round 1

Reviewer 1 Report

In my opinion, this paper is interesting and presented approach for definition of borehole-wall panorama can be used in engineering practice. Nevertheless, major revision is needed. The paper can be presented in more useful manner. My comments are the following:

English can be improved.

Dimensions of borehole-wall images presented in some figures can be increased.

Text of paper can be shortened. In particular, description of experimental procedure is excessively (!!!) long.

Number of equations can be shortened. The intermediate equations are not needed.

In my opinion, recent paper of Wang et al. (2017) can be mentioned in presented manuscript. Wang et al. (2017) have demonstrated the borehole television approach for detecting mining-induced fractures.

Wang X, Zhang D, Wang X, Zhang W (2017) Visual exploration of the spatiotemporal evolution law of overburden failure and mining-induced fractures: a case study of the Wangjialing coal mine in China. Minerals 7: 35

The comparison between technical specifications (CCD pixels, spatial precision, colour (in bits), etc. etc) of wide-angle camera used in this study and Borehole Digital Optical Televiewer (Li  et al.  2013) and introscopic camera (Skoczylas and Godyn 2014) can be useful.

Digital borehole camera and borehole digital optical televiewer  do not always have high technical specifications (CCD pixels, spatial precision, colour (in bits), etc.) and, therefore, fractures may be overlooked in case when filling material (debris of rock matrix) in fracture space and rock mass are of the same color (Palchik 2012, Bai and Tu 2019).  Are technical possibilities of novel approach  sufficient to  measure the width and length of fractures (Palchik 2012) when filling material (debris of rock matrix) in fracture space and rock matrix are of the same color ? 

Palchik V (2012) In situ study of intensity of weathering–induced fractures and methane emission to the atmosphere through these fractures. Engineering Geology 125: 56-65

Bai Q, Tu S (2019) A general review on longwall mining-induced fractures in near-face regions. Geofluids, Article ID 3089292, 22 pages

Conclusions must be more unequivocal. What is specific practical significance of obtained findings? It is important for the authors to clarify more clear which types, parameters and dimensions of fractures can be measured by using the proposed approach. It can be interesting for potential readers of applied journal.

Reviewer 2 Report

This is an interesting research

Author Response

Thank you for approving my job.

Reviewer 3 Report

The paper describes the use of image processing for the analysis of video sequences recorded in boreholes. The article is large and describes in detail the method developed by the Authors, which allows obtaining panoramic images from sequences taken in boreholes. I believe that the article may be interesting for researchers dealing with the analysis of such type of video sequences (for example for purposes of crack analysis, geological layers analysis, safety problems in mines, etc.).

Nevertheless, I think the Authors should respond to the following questions and objection:

1)

What were the technical aspects of the tested boreholes? What were their diameters and lengths? How long did it take to scan the borehole and what was the speed of the camera movement? In which rocks was the method tested?

1a)

What were the real sizes (in millimeters) of the analyzed images?

2)

How did the Authors deal with the problem of perspective? For photographs taken in the borehole, the distances in the 2D image do not always correspond to the 3D distance on the side faces of the holes (in the Z axis). This problem is not clearly described. On the other hand, from the presented results one can get the impression that the problem has been solved.

3)

Table 1 is not clear to me. It seems to me that the dc parameter should be an integer (this is suggested, for example, on Figure 24). In Table 1, however, these are floating-point numbers. Are these averages calculated from several measurements? Do the guidelines of formula 6 apply to the values in Table 1? (i.e., have all the pictures described in Table 1 been considered as invalid frames?)

Round 2

Reviewer 1 Report

In my opinion, paper was improved. Nevertheless, moderate revision is needed. My comments are the following:

The comparison between technical specifications (CCD pixels, spatial precision, colour (in bits), etc. etc) of wide-angle camera used in this study and Borehole Digital Optical Televiewer (Li et al. 2013) and introscopic camera (Skoczylas and Godyn 2014) is not presented. Why ? The author’s response [“With our method, the forward-looking borehole video can be converted to the corresponding borehole-wall cylindrical panorama. Then engineers can analyze the borehole information quantitatively using the cylindrical panorama, including features such as the distribution and width of a fracture, the thickness and inclination of a coal seam, the strike, dip, dipping direction and developing degree of the structure plane”] is not convincing (!!!). Unfortunately, the authors try to avoid the study of lengths, width and inclination of different fractures (and etc. and etc.) in this paper.. Why do the authors hope that other unknown engineers can resolve these very important problems instead of them ? (J). Please add few sentences (to text of the paper !!!) about case where filling material in fracture space and rock matrix are of the same colour since this point is important for detecting of any fractures in sides of boreholes and shafts, and in close-to-vertical faults. Please add the paper of Palchik (2012) to text of revised paper.  Palchik V (2012) In situ study of intensity of weathering–induced fractures and methane emission to the atmosphere through these fractures. Engineering Geology 125: 56-65

Reviewer 3 Report

The Authors have made the changes I suggest and I think that the paper can be published in Applied Sciences.

Author Response

Thank you for approving my job.